# AN ALGORITHM TO PERFORM COVARIANCE-ADJUSTED SUPPORT VECTOR CLASSIFICATION IN NON-EUCLIDEAN SPACES

## ABSTRACT

Traditional Support Vector Machine (SVM) classification is carried out by finding the max-margin classifier for the training data that divides the margin space into two equal sub-spaces. This study demonstrates limitations of performing Support Vector Classification in non-Euclidean spaces by establishing that the underlying principle of max-margin classification and Karush Kuhn Tucker (KKT) boundary conditions are valid only in the Euclidean vector spaces, while in non-Euclidean spaces the principle of maximum margin is a function of intra-class data covariance. The study establishes a methodology to perform Support Vector Classification in Non-Euclidean Spaces by incorporating data covariance into the optimization problem using the transformation matrix obtained from Cholesky Decomposition of respective class covariance matrices, and shows that the resulting classifier obtained separates the margin space in ratio of respective class population covariance. The study proposes an algorithm to iteratively estimate the population covariance-adjusted SVM classifier in non-Euclidean space from sample covariance matrices of the training data. The effectiveness of this SVM classification approach is demonstrated by applying the classifier on multiple datasets and comparing the performance with traditional SVM kernels and whitening algorithms. The Cholesky-SVM model shows marked improvement in the accuracy, precision, F1 scores and ROC performance compared to linear and other kernel SVMs.

## 1 INTRODUCTION

Support Vector Machines (SVM), derived from Vapnik's statistical learning theory( Vapnik (2013)) is a powerful kernel-based machine learning tool that is suitable for both classification and regression tasks, and hence is widely used in diverse fields ranging from pattern recognition( Byun & Lee (2002)) to text classification( Isa et al. (2008)), image classification( Mustafa et al. (2017)) and forecasting in finance( Ince (2000)). Unlike other classification algorithms that focus on empirical risk minimization, SVM focuses on structural risk minimization( Burges (1998)),( Scholkopf et al. (1999)). It achieves this by identifying a separating linear hyperplane with maximum margin from the margin-edge hyperplanes that bound the two data classes in the N-dimensional space, N being the number of features. The parameters of the margin-edge hyperplanes and the decision boundary are estimated by constructing an optimization problem and solving it using quadratic programming. If the data are not linearly separable, kernel tricks are applied where the kernels transform the data from input space to a higher-dimensional feature space where they can be linearly separable( Schölkopf & Smola (2002)).

The equation of the linear SVM hyperplanes for both the decision boundary and margin hyperplanes, and the margin calculations, are derived from the Euclidean distance formula. However,P.C. Mahalanobis( Mahalanobis (2018)) while proposing Mahalanobis distance showed that in the input space (also called the statistical space in this study) where the data points are present, the distance between two data points is to be measured by statistical distance (accounting for data covariance) and not the Euclidean distance. Hence, traditional SVM, which is built on foundations of Euclidean distance, should not be valid in the input space as it is Non-Euclidean. Secondly, the concept of Max-Margin Classification considers the decision boundary to be equidistant from either margin

hyperplane. SVM optimization problem uses KKT boundary conditions, where only the support vectors at the margin boundaries play a role in deciding the decision boundary. The role of the other data points, as well as information about the data class distributions and inherent variance-covariance structure of the data plays a minimal role( Zafeiriou et al. (2007)). However, a separating hyperplane should divide the margin space in ratio of the data dispersion of each class- a higher margin should be provided for the data class having higher dispersion (higher variance) and lower margins for the data class having lower dispersion (lower variance) as it is more cohesive and compact. The entire dataset(through variance) should influence the calculation of decision boundary. Hence, adjustments need to be made for class distribution variance while calculating and maximizing the margins in the optimization problem.

Several studies have acknowledged the aforementioned issue and have recommended various approaches to incorporate variance into the SVM optimization problem. ( Tsang et al. (2006)) incorporated covariance information in one-class SVMs by using the Mahalanobis distance instead of Euclidean distance to calculate the margin. ( Peng & Xu (2012)) incorporated the Mahalanobis distance into the twin support vector machine (TSVM) to determine two optimization problems to determine the two non-parallel separate hyperplanes. ( Ke et al. (2018)) presented a Mahalanobis distance-based biased least squares support vector machine (MD-BLSSVM) to classify PU data. ( Huang et al. (2004)) proposed the maxi-min margin machine that incorporates class distribution information into decision boundary optimization problem using statistical distance. ( Wang et al. (2007)) proposed weighted Mahalanobis distance Kernels for SVMs that incorporates covariance information into existing kernels. ( Zafeiriou et al. (2007)) proposed the minimum class variance SVMs (MCVSVMs) by optimizing Fisher's Discriminant Analysis where a within-class scatter matrix is incorporated into optimization problem to account for the data variance. However, analysis of the optimization problems formulated in those studies revealed gaps in application of appropriate vector spaces and dimensional inconsistencies. In this study we have tried to rectify those gaps by proposing Covariance-Adjusted Support Vector Machine (CSVM)-building upon the concepts that (1) statistical space is different from the Euclidean space, (2) SVM is valid in Euclidean space only as it is derived using Euclidean distance, and (3) Mahalanobis distance is essentially a transformation of the data from the statistical space to the Euclidean space ( Sahoo & Maiti (2025)). Using these concepts, we first transform data from Non-Euclidean statistical space to Euclidean space by performing Cholesky Decomposition of intra-class covariance matrix for each data class, and using the lower triangular matrix as transformation matrix. Then we formulate the SVM optimization problem in the Euclidean space, and reverse-transform it to the original input space. Here, the Cholesky decomposed lower-triangular matrix of the covariance matrix transforms the data from statistical space to Euclidean space, thus mirroring the kernel trick. While using whitening algorithms (Cholesky Decomposition or Eigen Decomposition) to transform data to Euclidean space and performing SVM is standard practice these days, we have tried two new things: First, in this study we have examined the effects of whitening on SVM optimization problem formulation in Non-Euclidean Space. By doing reverse whitening and transforming SVM optimization problem in Non-Euclidean space, we see that (1) an N-class support-vector classification problem results in N decision boundaries in the input space; (2) equivalence between statistical space and the Euclidean space implies that the decision boundary should split the margin between the margin boundaries in the input space in ratio of respective data class covariances; (3) KKT Boundary conditions are not valid while attempting SVM in a Non-Euclidean Space. Second, we address the limitations of calculating population covariance matrix in absence of test data label information by proposing the SM Algorithm- that iteratively calculates the population covariance matrix and performs Covariance-adjusted SVM (CSVM) from training data information. Finally, effectiveness of CSVM is demonstrated by applying on 5 standard datasets and comparing the results with traditional linear and other kernel SVMs, as well as with other prevalent whitening methods. CSVM shows better classification table metrics and higher AUC values in ROC curves compared to other models.

The remainder of this paper is organized as follows: Section 2 presents a mathematical derivation of the vector space transformation from statistical to Euclidean using Cholesky decomposition and formulates the optimization problem of the SVM classifier after adjusting for data covariance in the Euclidean space. Section 3 gives the SM Algorithm to iterately calculate population covariance from training data sample covariance. Comparison with other studies done in literature and other whitening approaches is done in section 4. Section 5 gives the results of application of CSVM and other SVM approaches on 5 datasets and performance is compared using precision, recall, accuracy,

F1 scores and ROC curves. Finally, sections 6 concludes with a summary of the contributions, limitations and future scope of work.

## 2 VECTOR SPACE TRANSFORMATION TO EUCLIDEAN SPACE FOR SVM

For a probability distribution of a population, distance between a point and the mean in statistical/input space given by the Mahalanobis distance written as:

$$(X-\mu)^T \Sigma^{-1}(X-\mu) = \left[\Psi^{-1}(X-\mu)\right]^T \left[\Psi^{-1}(X-\mu)\right] = \left[\Psi^{-1}X - \Psi^{-1}\mu\right]^T \left[\Psi^{-1}X - \Psi^{-1}\mu\right] \tag{1}$$

where $\Sigma$ is the population covariance matrix and $\Sigma = \Psi\Psi^T$ is the Cholesky decomposition of $\Sigma$ resulting in the lower triangular matrix $\Psi$. While the LHS in (1) represents the distance between raw data in statistical/input space, its equivalence in the RHS is the formula for Euclidean distance between the data points that are transformed to a new vector space using transformation matrix $\Psi^{-1}$, and represented by the inner product $\left[\Psi^{-1}(X-\mu)\right]^T \left[\Psi^{-1}(X-\mu)\right]$. The new vector space has following properties:

- The space has an inner product, hence it is an inner product space.
- The inner product represents the distance between the points.
- If $\mu = 0$, the inner product represents the norm.

All these are properties of Euclidean space. Hence the new vector space is the Euclidean Space with the transformed data is given by

$$X^{\text{Euclidean}} = \Psi^{-1} X^{\text{Input}} \tag{2}$$

and the original statistical/input space is a non-Euclidean space.

Mahalanobis transformation uses the covariance matrix of the underlying probability distribution, which is derived from the population/sample. In SVM of binary classification, the dataset belongs to two classes given by $y = 1$ or $y = -1$. As these are two distinct labels with distinct features and parameters, they can be considered as two distinct populations, which correspond to two distinct distributions. These distributions have their unique covariance structure, hence will be characterized by their distinct covariance matrices $\Sigma_{y=1}$ and $\Sigma_{y=-1}$ respectively. Accordingly, the data transformation is distinct for each label, given by

$$X_{y=1}^{\text{Euclidean}} = \Psi_{y=1}^{-1} X_{y=1}^{\text{Input}} \qquad \text{and} \qquad X_{y=-1}^{\text{Euclidean}} = \Psi_{y=-1}^{-1} X_{y=-1}^{\text{Input}} \tag{3}$$

In SVM, the objective is to find a linear hyperplane that separates the data with maximum margin from the data points of either class. The equation of the hyperplane and the magnitude of the margin are derived from the Euclidean distance formula and principles of the Cartesian coordinate system. Since these operations are valid in the Euclidean space only, the following lemma is proposed:

***Lemma 2.1. Principles of support vector classification (KKT boundary conditions and max-margin classification) are valid only when the data is transformed from the input/statistical space to the Euclidean space using $\Psi^{-1}$, which is inverse of the lower triangular matrix obtained by Cholesky Decomposition of the population covariance matrix. The SVM optimization problem, loss function and constraints are formulated in the transformed Euclidean space.***

Performing linear SVM the Euclidean space results in the equation of the maximum margin classifier (Considering hard margin SVM, $\xi_i = 0$):

$$\theta^T X^{\text{Euclidean}} + \theta_0 = 0 \tag{4}$$

$$\text{With the margin value} \qquad \frac{1}{\sqrt{\theta^T \theta}} \tag{5}$$

So the optimization problem becomes:

$$\min \frac{1}{2}\theta^T \theta \tag{6}$$

$$\text{Subject to} \qquad y_i(\theta^T X^{\text{Euclidean}} + \theta_0) \geq 1 \tag{7}$$

Formulating Euclidean space-Non Euclidean input space equivalence, the classifier given in (4) for data labelled $y = 1$ becomes (Considering hard margin SVM, $\xi_i = 0$)

$$\theta^T \Psi_{y=1}^{-1} X_{y=1}^{\text{Input}} + \theta_0 = 0 \qquad or \qquad \left((\Psi_{y=1}^{-1})^T \theta\right)^T X_{y=1}^{\text{Input}} + \theta_0 = 0 \tag{8}$$

What the margin of the linear classifier hyperplane given in (8) in the input space? If hypothetically, we apply the rules of Cartesian coordinate geometry and try to calculate the margin from the margin boundary to the classifier, the margin is, the margin value for $y = 1$ is

$$\frac{1}{\sqrt{\left((\psi_{y=1}^{-1})^T \theta\right)^T \left((\psi_{y=1}^{-1})^T \theta\right)}} = \frac{1}{\sqrt{\theta^T (\psi_{y=1}^T \psi_{y=1})^{-1} \theta}} = \frac{1}{\sqrt{\theta^T (\Sigma_{y=1})^{-1} \theta}} \tag{9}$$

Since $\Sigma^T = \Sigma$ as $\Sigma$ is a symmetric matrix.

Hence, the margin maximization problem in the input space for $y = 1$ becomes

$$\text{Minimize} \quad \frac{1}{2} \theta^T (\Sigma_{y=1})^{-1} \theta \tag{10}$$

$$\text{Satisfying the constraint} \qquad \theta^T \Psi_{y=1}^{-1} X_{y=1}^{\text{Input}} + \theta_0 \geq 1 \tag{11}$$

Similarly, for $y = -1$ the optimization problem in the input space becomes

$$\text{Minimize} \quad \frac{1}{2} \theta^T (\Sigma_{y=-1})^{-1} \theta \tag{12}$$

$$\text{Satisfying the constraint} \qquad \theta^T \Psi_{y=-1}^{-1} X_{y=-1}^{\text{Input}} + \theta_0 \leq -1 \tag{13}$$

There are two sets of optimization problems to be solved. This gives rise to the following lemma:

***Lemma 2.2. For a two-class problem, the application of SVM in the input space domain generates not one, but two unique optimization problem formulations resulting in two unique linear classifiers—each input space having its own linear classifier. In an N-class problem, there will be N data class distributions and N input spaces; hence, there will be N linear classifiers.***

Calculating margin for $y = 1$ using equation (9) and similarly calculating margin for $y = -1$ and comparing them results in

$$\frac{\text{Margin}_{y=1}}{\text{Margin}_{y=-1}} = \frac{\sqrt{\theta^T (\Sigma_{y=-1})^{-1} \theta}}{\sqrt{\theta^T (\Sigma_{y=1})^{-1} \theta}} \tag{14}$$

The following lemma is proposed:

***Lemma 2.3. In the input space, margin for each data class is a function of respective population covariance matrix, given by $1/\Sigma^{-1}$. Hence the assumptions of KKT boundary conditions are not valid in the input space as each data point (other than support vectors) contributes to $\Sigma^{-1}$.***

Hence unlike the original support vector machine algorithm, in a Non-Euclidean space the margin of the classifier is dependent on the intra-class data covariance (variance is essentially covariance of a random variable with itself).

## 3 ALGORITHM FOR SVM CLASSIFICATION FROM SAMPLE COVARIANCE INFORMATION

CSVM vector transformation requires knowing the population covariance matrix $\Sigma$, which is not possible as the test data labels are unknown. Here we propose an algorithm here to perform Covariance-Adjusted SVM classification using the sample covariance matrices $S_{y=1}$ and $S_{y=-1}$ that can be calculated from training data (We call it the **SM Algorithm**). We have established that if we do SVM in the Euclidean space, the corresponding linear classifier in the input space divides the margin as a function of $1/\Sigma^{-1}$ as given in equation (9), and it can give an indicative location

of the test data w.r.t the classifier and its labels. Hence in this algorithm we iteratively perform the following steps: (1) Calculate the covariance matrices for the available labelled data (starting with training data) to obtain $S_{y=1}$ and $S_{y=-1}$; (2) Find the SVC linear classifier that divides the margin in the ratio $\sqrt{\frac{\theta^T (\Sigma_{y=-1})^{-1} \theta}{\theta^T (\Sigma_{y=1})^{-1} \theta}}$ , and identify the test data on either side of the modified classifier to get the new labelled datasets. The steps are repeated till convergence is attained.

Below are the steps for SM algorithm:

1. Initialization phase:
   (a) Name original training data labelled 1 and $-1$ as $\text{Train}_1$ and $\text{Train}_{-1}$ respectively.
   (b) Calculate covariance matrices for $\text{Train}_1$ and $\text{Train}_{-1}$ i.e. $S_{y=1}$ and $S_{y=-1}$ respectively.

2. Iteration phase:
   (a) Perform Cholesky Decomposition of covariance matrices to obtain $C_{y=1}$ and $C_{y=-1}$, where
   $$S_{y=1} = C_{y=1} C_{y=1}^T \quad \text{and} \quad S_{y=-1} = C_{y=-1} C_{y=-1}^T.$$
   (b) Calculate $C_{y=1}^{-1} \text{Train}_1$ and $C_{y=-1}^{-1} \text{Train}_{-1}$ to transform $\text{Train}_1$ and $\text{Train}_{-1}$ data from input space to Euclidean space.
   (c) Perform support vector classification on $\text{Train}_1$ and $\text{Train}_{-1}$ data in the Euclidean space and calculate the parameter vector $\theta_{\text{Euclidean}}$.
   (d) Perform linear SVM on the original $\text{Train}_1$ and $\text{Train}_{-1}$ data (identified in initialization phase) in the input space and calculate the equation of the linear classifier
   $$\theta_{\text{Input}}^T x + \theta_0 = 0.$$
   (e) Adjust $\theta_0$ to $\theta_0'$ and modify the classifier calculated in step (d) so that the modified classifier
   $$\theta_{\text{Input}}^T x + \theta_0' = 0$$
   divides the margin in the input space in ratio
   $$\sqrt{\frac{\theta_{\text{Euclidean}}^T (S_{y=-1})^{-1} \theta_{\text{Euclidean}}}{\theta_{\text{Euclidean}}^T (S_{y=1})^{-1} \theta_{\text{Euclidean}}}}.$$
   (f) Label test datapoints as $+1$ and $-1$ based on whether
   $$\theta_{\text{Input}}^T X_{\text{Test}} + \theta_0' \geq 0 \quad \text{or} \quad \leq 0.$$
   (g) Add the test datapoints to $\text{Train}_1$ and $\text{Train}_{-1}$ based on their labels identified in step (f) to obtain updated $\text{Train}_1$ and $\text{Train}_{-1}$ data.
   (h) Re-calculate covariance matrices for new $\text{Train}_1$ and $\text{Train}_{-1}$ to get new $S_{y=1}$ and $S_{y=-1}$ respectively.
   (i) Repeat steps (a) to (h) until convergence.

3. Convergence criteria:
   (a) Check if the test data assignments have stopped changing have stopped moving (i.e., the changes in test data labels are below a certain threshold).
   (b) If test data has converged, terminate the algorithm.

   End Algorithm

## 4 COMPARISON OF CSVM WITH EXISTING WORK AND WHITENING ALGORITHMS

The CSVM model and algorithm proposed essentially transforms the data into Euclidean space by decorrelating and standardizing the data and then performs support vector classification in that space. This process is akin to data whitening. Data whitening has become a standard data pre-processing technique with many whitening algorithms being widely used e.g. PCA Whitening, ZCA Whitening etc. Apart from studying effects of whitening on Non-Euclidean SVM, other novelties of this study are:

- While data whitening leads to superior performance of support vector machines in many cases, the literature is divided on the reasons for this. In this study we provide a vector space explanation of why whitening works: *whitening transforms the data from non-Euclidean space/input space to Euclidean space, and it is in Euclidean space that the equations for various ML models like support vector machines, KNN, K-Means clustering are based.*

- PCA/ZCA whitening algorithms perform whitening of the entire training data and then apply them on the test data. However, in many cases the data corresponding to various labels belongs to different populations and can have stark differences in their features. Hence whitening of the data needs to be carried out separately class-wise, as is done in this study.

- Instead of applying training data whitening transformation on test data, SM algorithm iteratively iteratively tries to identify labels of test data and thus calculate the population covariance matrix, leading to better performance vis-a-vis PCA and ZCA Whitening.

- By starting from first principles and focusing on Mahalanobis Distance as a vector space transformation, we formulate the optimization problem in CSVM keeping it both vector space and dimensionally consistent, and thus address the limitations of previous studies done in variance adjusted SVM.

## 5 RESULTS AND DISCUSSION

For experimental verification of covariance-adjusted SVM (CSVM) as well as the effectiveness of SM algorithm, the CSVM model was applied on five popular datasets- Breast Cancer Wisconsin Dataset, OSHA Dataset, Diabetes Dataset, Red Wine Dataset and Pulsar Dataset. These datasets are binary class and the objective of selecting the 5 datasets was to check for generalizability of the model in various domains- healthcare, astronomy, quality, and safety/text mining. First the dataset was split into training and validation data in the ratio 80:20. Then CSVM model was applied on the data and the classification table giving the accuracy, precision, recall and F1 scores and Receiver Operating Characteristic (ROC) curves giving the Area Under Curve (AUC) were obtained. In addition, the effectiveness of CSVM model is compared with other widely used support vector classification models by applying various support vector kernels - linear, RBF, Sigmoid and Polynomial- on the datasets and obtaining the classification tables and ROC curves. The accuracy, precision, recall, F1 scores and AUC values were compared.

In addition, owing to similarity with whitening algorithms, we compared CSVM with PCA and ZCA whitening cum SVM approach- PCA and ZCA whitening were carried out on the 5 datasets and then linear SVM was used and classification table and ROC curves were obtained. The accuracy, precision, recall, F1 scores and AUC values were then compared for our model, PCA whitening-linear SVC and ZCA whitening-linear SVC.

The accuracy, precision, recall and F1 scores of the SVM kernels and whitening models over the 5 datasets is given in Table 1, 2, 3 and 4 respectively. It can be seen that our model CSVM-Cholesky Decomposition attains highest accuracy, recall and F1 scores for 4 datasets-Breast Cancer, Pulsar, Red wine and Diabetes and in the OSHA dataset it has the second highest precision and performs better than linear SVM. CSVM-Cholesky Decomposition attains highest precision for 3 datasets- Breast cancer, diabetes and red wine, and in the OSHA dataset performs better than linear SVM while in Pulsar dataset, linear SVM performs better.

Finally, the performance comparison of the SVM kernels and PCA/ZCA whitening methods is compared by drawing the ROC curves and measuring the Area Under Curve (AUC) values. It can be seen that for Breast Cancer, Pulsar and Red Wine datasets, our CSVM-Cholesky transformation model is having the highest AUC values. For OSHA data, CSVM AUC is joint highest with RBF Kernel and CSVM is joint highest with linear, PCA and ZCA values.

Table 1: Comparison of accuracy of SVM kernels and whitening models over 5 datasets

| Method | Breast Cancer | OSHA | Diabetes | Red Wine | Pulsar |
|---|---|---|---|---|---|
| CSVM–Cholesky | **0.974** | 0.752 | **0.786** | **0.744** | **0.981** |
| SVM–Linear | 0.956 | 0.741 | 0.760 | 0.731 | 0.979 |
| SVM–RBF | 0.947 | **0.760** | 0.760 | 0.650 | 0.973 |
| SVM–Sigmoid | 0.465 | 0.671 | 0.656 | 0.422 | 0.925 |
| SVM–Poly | 0.947 | 0.747 | 0.760 | 0.637 | 0.971 |
| SVM Linear–PCA | 0.947 | 0.741 | 0.760 | 0.728 | 0.979 |
| SVM Linear–ZCA | 0.939 | 0.741 | 0.760 | 0.738 | 0.979 |

Table 2: Comparison of precision of SVM kernels and whitening models over 5 datasets

| Method | Breast Cancer | OSHA | Diabetes | Red Wine | Pulsar |
|---|---|---|---|---|---|
| CSVM–Cholesky | **0.974** | 0.747 | **0.775** | **0.744** | 0.954 |
| SVM–Linear | 0.96 | 0.742 | 0.738 | 0.733 | **0.962** |
| SVM–RBF | 0.961 | **0.766** | 0.743 | 0.674 | 0.954 |
| SVM–Sigmoid | 0.380 | 0.655 | 0.633 | 0.417 | 0.770 |
| SVM–Poly | 0.961 | 0.758 | 0.775 | 0.696 | 0.939 |
| SVM Linear–PCA | 0.948 | 0.742 | 0.738 | 0.731 | 0.962 |
| SVM Linear–ZCA | 0.937 | 0.742 | 0.738 | 0.739 | 0.962 |

Table 3: Comparison of recall of SVM kernels and whitening models over 5 datasets

| Method | Breast Cancer | OSHA | Diabetes | Red Wine | Pulsar |
|---|---|---|---|---|---|
| CSVM–Cholesky | **0.970** | 0.721 | **0.744** | **0.748** | **0.925** |
| SVM–Linear | 0.946 | 0.704 | 0.736 | 0.736 | 0.906 |
| SVM–RBF | 0.930 | **0.723** | 0.716 | 0.617 | 0.876 |
| SVM–Sigmoid | 0.401 | 0.653 | 0.639 | 0.416 | 0.762 |
| SVM–Poly | 0.930 | 0.705 | 0.688 | 0.596 | 0.876 |
| SVM Linear–PCA | 0.939 | 0.704 | 0.736 | 0.734 | 0.906 |
| SVM Linear–ZCA | 0.932 | 0.704 | 0.736 | 0.742 | 0.906 |

Table 4: Comparison of F1 Scores of SVM kernels and whitening models over 5 datasets

| Method | Breast Cancer | OSHA | Diabetes | Red Wine | Pulsar |
|---|---|---|---|---|---|
| CSVM–Cholesky | **0.972** | 0.728 | **0.754** | **0.743** | **0.939** |
| SVM–Linear | 0.953 | 0.711 | 0.737 | 0.731 | 0.932 |
| SVM–RBF | 0.942 | **0.731** | 0.725 | 0.601 | 0.910 |
| SVM–Sigmoid | 0.385 | 0.654 | 0.635 | 0.416 | 0.766 |
| SVM–Poly | 0.942 | 0.712 | 0.699 | 0.560 | 0.904 |
| SVM Linear–PCA | 0.943 | 0.711 | 0.737 | 0.728 | 0.932 |
| SVM Linear–ZCA | 0.934 | 0.711 | 0.737 | 0.737 | 0.932 |

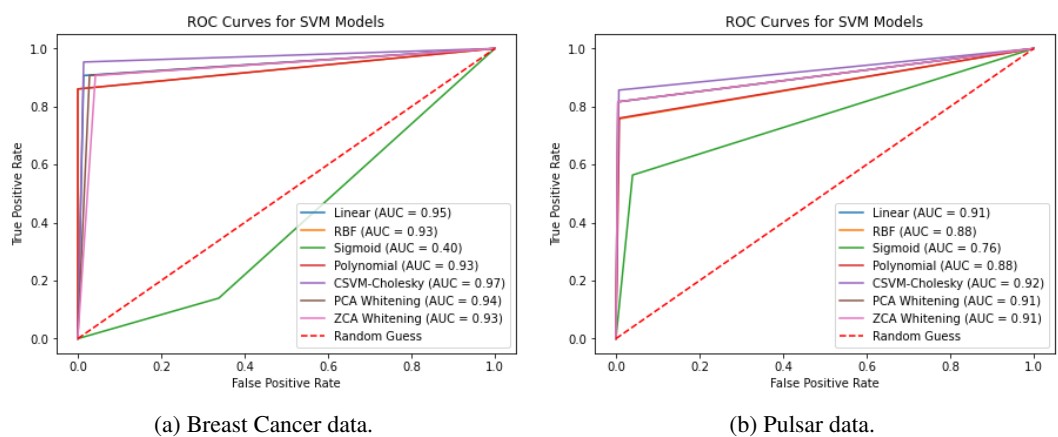

(a) Breast Cancer data.

(b) Pulsar data.

Figure 1: ROC Curves and AUC values for SVM kernels and whitening models for Breast Cancer and Pulsar data.

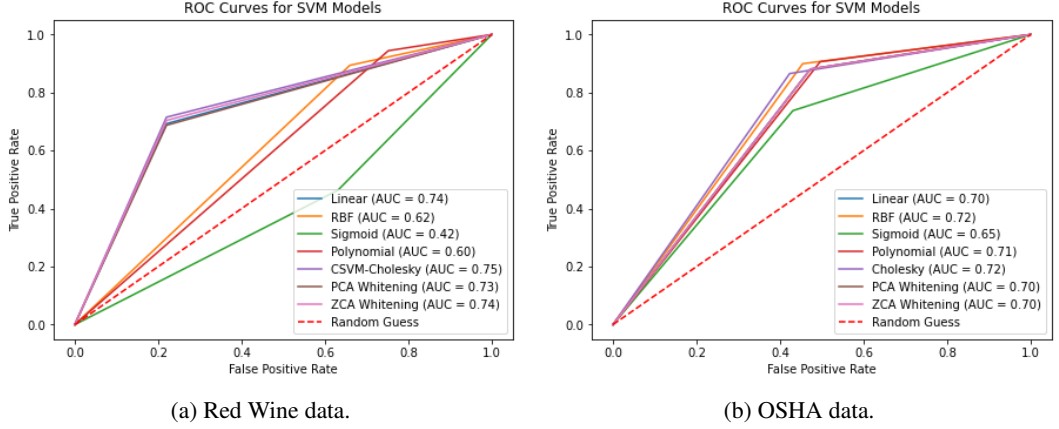

(a) Red Wine data.

(b) OSHA data.

Figure 2: ROC Curves and AUC values for SVM kernels and whitening models for Red Wine and OSHA data.

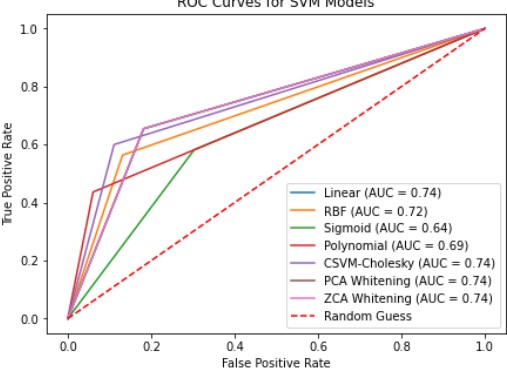

Figure 3: ROC Curves and AUC values for SVM kernels and whitening models for Diabetes data.

## 6 Conclusion, limitations, and future work

It can be seen that the CSVM model is a powerful tool that gives better results compared to traditional SVM kernels. The purpose of experimenting and comparing with traditional SVM approaches over 5 datasets was to validate the findings of lemma 2.1, 2.2 and 2.3. Hence in this study we demonstrate that (1) SVM optimization problem formulation and solutions are valid in the Euclidean space, and formulating and solving them in the input space carries risk of misclassification; (2) A binary class problem requires one classifier in the Euclidean space but two classifiers in the Non-Euclidean space, while an N class problem has one classifier in Euclidean Space but N classifiers in the Non-Euclidean input space; (3) The distance of the classifiers from their respective margin boundaries is a function of their intra-class covariances.

Despite the performance of Cholesky transformation-CSVM, it suffers from certain drawbacks. First, it requires the knowledge of the population covariance structure of the data distribution. In absence of information about population covariance, the transformation of data to Euclidean space becomes difficult. While SM algorithm has been proposed to address this limitation, it is a heuristic algorithm and though it gives the best classification performance, perfect classification is yet to be achieved. Secondly, the computational complexity of Cholesky kernel is higher than traditional linear SVM as extra steps of calculating covariance matrices and Cholesky decomposition are involved. Hence it leads to the following dilemma: is the increase in classification performance worth the computational complexity?

Considering the limitations, future work will involve finetuning the SM algorithm so that we can reduce the computational complexity.

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

## A  APPENDIX

You may include other additional sections here.

