# OpenReview forum: "An Algorithm to perform Covariance-Adjusted Support Vector Classification in Non-Euclidean Spaces"
_ICLR.cc/2026/Conference — Submitted to ICLR 2026_

### Official Review · Reviewer_ZbFi · 2025-10-26

**Soundness:** 2
**Presentation:** 2
**Contribution:** 2
**Rating:** 2
**Confidence:** 4

**Summary:**

The study highlights limitations of traditional Support Vector Machine (SVM) classification in non-Euclidean spaces. It introduces a method for SVM classification in non-Euclidean spaces by incorporating data covariance through a transformation matrix derived from Cholesky decomposition of class covariance matrices. This approach separates the margin space proportionally to class population covariance. The proposed algorithm iteratively estimates a covariance-adjusted SVM classifier from sample covariance matrices.

**Strengths:**

The study establishes a model for SVM classification in non-Euclidean spaces.

The proposed algorithm iteratively estimates the population covariance-adjusted SVM classifier from sample covariance matrices.

 Empirical results demonstrate significant improvements over traditional SVM methods.

**Weaknesses:**

The method requires computing covariance matrices for each class and performing Cholesky decomposition.

 The performance of Cholesky-SVM depends heavily on the accuracy of the estimated covariance matrices. If sample sizes are small or the data is noisy, covariance estimates may be unreliable.

 Computing the inverse of matrices may be time-consuming for high-dimensional data.

 The paper does not explicitly mention the computational complexity of the model.

SVMs in the hyperbolic space have been proposed.

 The experiments on real data sets are limited.

**Questions:**

See weaknesses.

---

### Official Review · Reviewer_EArk · 2025-10-28

**Soundness:** 2
**Presentation:** 3
**Contribution:** 1
**Rating:** 2
**Confidence:** 2

**Summary:**

This paper presents a novel approach to Support Vector Machine (SVM) classification, named Covariance-Adjusted SVM (CSVM), designed specifically for non-Euclidean spaces. The authors' core argument is that traditional SVMs, which are based on maximizing the margin using Euclidean distance, are theoretically flawed when applied to statistical input spaces where data covariance is significant. They posit that in such spaces, the principle of maximum margin is a function of the intra-class covariance.

The proposed solution involves a two-step process. First, the data for each class is transformed from the non-Euclidean input space to a true Euclidean space. This is achieved using a transformation matrix derived from the Cholesky Decomposition of each class's covariance matrix. Second, a standard SVM is formulated and solved in this transformed space. When mapped back to the original input space, this method results in a classifier that divides the margin in a ratio proportional to the respective class covariances.

**Strengths:**

1.  **Strong Theoretical Foundation:** The paper is built on a clear and compelling theoretical argument. The distinction between the non-Euclidean statistical space and the Euclidean vector space is well-articulated, providing a strong justification for why traditional SVMs might be suboptimal. Grounding the methodology in the Mahalanobis distance and vector space transformations gives the work significant theoretical weight.
2.  **Practical Algorithm for an Unseen Data Problem:** The SM algorithm is a clever and practical solution to the real-world problem of unknown population covariance. By iteratively refining the covariance estimate using the test data, it moves beyond a static transformation based solely on training data, which likely contributes to its superior performance.

**Weaknesses:**

1.  **Computational Complexity and Scalability:** The iterative nature of the SM algorithm appears computationally expensive. It requires recalculating covariance matrices and their decompositions, as well as re-labeling the entire test set at each step. The paper acknowledges this but does not provide a formal analysis of the computational complexity or discuss the method's feasibility for very large-scale datasets.
2.  **Convergence Properties of the SM Algorithm:** The SM algorithm is presented as a heuristic. The paper does not discuss its convergence properties, such as whether convergence is guaranteed or if the algorithm could be sensitive to initialization (e.g., the initial training/test split) and potentially settle in a poor local minimum.
3.  **Limited Scope of Experimental Comparison:** The evaluation, while thorough within the SVM family, is somewhat insular. The paper does not compare CSVM against other powerful, non-SVM classification paradigms like Multi-Layer Perceptrons (MLPs) or Gradient Boosting. This makes it difficult to assess the method's competitiveness in a broader machine learning context, particularly on more complex datasets where these alternative models are often more suitable. **If the authors could add experiments comparing CSVM to other types of classifiers, I would be inclined to increase my rating of this paper.**

**Questions:**

1.  Could you elaborate on the convergence properties of the SM algorithm? Is convergence guaranteed, and how sensitive is its performance to the initial state (i.e., the training data)?
2.  The Cholesky decomposition requires the covariance matrix to be positive definite and invertible. How does your method handle high-dimensional datasets where the number of features exceeds the number of samples, leading to a singular sample covariance matrix?
3.  The paper makes the interesting claim that an N-class problem would result in N classifiers in the input space. Have you considered extending this framework to multi-class problems, and what challenges do you foresee in adapting the SM algorithm for that scenario?
4.  **Performance Against Non-SVM Classifiers:** Your experiments convincingly show CSVM's superiority over other SVM variants. However, since SVM is a classifier, a comparison to other state-of-the-art classifiers is essential. How do you expect CSVM to perform against fundamentally different models, such as an MLP, especially on more complex and larger-scale datasets? Including such a baseline would provide valuable context for the performance improvements observed. As mentioned, **adding these experiments would significantly strengthen my overall evaluation of this work.**

---

### Official Review · Reviewer_vm1i · 2025-11-02

**Soundness:** 2
**Presentation:** 1
**Contribution:** 2
**Rating:** 2
**Confidence:** 3

**Summary:**

This paper is based on the observation that while the SVM algorithm assumes datapoints live in the usual Euclidean space ($\mathbb{R}^{n}$ equipped with the Euclidean metric), the Mahalanobis distance is the more natural metric between random vectors in a dataset. The authors note that the Euclidean geometry assumption is necessary for SVM to be valid, and conclude that a different algorithm is needed. They develop one by first using the population covariance matrix to transform data points back into a Euclidean representation, which can be directly subjected to SVM. An algorithm for estimating the population covariance from data covariance is also provided. Comparing their work against several baselines on 5 datasets, the authors report favorable results on 4 of them.

**Strengths:**

* The authors' contribution continues a pre-existing line of research, using the Mahalanobis distance to make SVMs flexible and robust in non-Euclidean spaces.

* The proposed algorithm seems to be intuitive and obvious, and empirically verified.

* The proposed algorithm produces better results compared to traditional SVM with widely used kernels in four out of five datasets.

**Weaknesses:**

* The idea is straightforward and obvious. There is a vast literature on metric learning in machine learning.

* The paper does not discuss or compare with more latest kernel functions, such as data-dependent kernels.

Kai Ming Ting, Yue Zhu, and Zhi-Hua Zhou (2018). Isolation Kernel and Its Effect on SVM. In Proceedings of the 24th ACM SIGKDD.

V. V. Malgi, S. Aryal, Z. Rasool, and D Tay (2023). Data-dependent and Scale-Invariant Kernel for Support Vector Machine Classification. In: Proceedings of the PAKDD 2023.

* Experiments are too slim, used only five small datasets.

* None of the stated Lemmas are proven, and, furthermore, their formal statements are a bit vague.

* The presentation is not clear to the reader at several places. For example: (i) In section 2, after transforming both positive and negative instances, the introduction of the margin ratio is a bit hard to follow (recommend reformulating the section starting at equation 8). (ii) What does the acronym SM stand for? (iii) The introduced algorithm is referred to as Covariance-adjusted SVM in the body of the paper and as Cholesky-SVM in the abstract, which is confusing.

**Questions:**

I would appreciate the author's comments on the weaknesses raised above.

---

### Official Review · Reviewer_xevi · 2025-11-03

**Soundness:** 2
**Presentation:** 2
**Contribution:** 2
**Rating:** 2
**Confidence:** 5

**Summary:**

This paper proposes a covariance-adjusted support vector machine. The main idea is to transform (or whiten) each class of data via Cholesky decompostion of the sample class covariance, which follows a linear SVM in the transformed Euclidean space. The bias term is then adjusted to maintain the ratio of margins in proportion to the class covariance. An interative SM algorithm is also developed to leverage the information from test data to refine the estimation of the class covariance. Across five benchmark binary datasets, the proposed method shows higher accuracy, precision, recall, F1, and ROC‑AUC than baseline methods including linear and kernel SVMs.

**Strengths:**

1. This paper studies an interesting and important question, as SVM is a fundamental method in machine learning with strong theoretical guarantees and solid empirical performance on tabular data, where deep learning is often outperformed by SVMs and ensemble methods.

2. The empirical results are presented as strong: the authors claim the proposed method outperforms key baselines.

**Weaknesses:**

1. The main claim is that “SVM is valid in Euclidean space only.” Kernel SVM is well established and operates via kernels in implicit feature (Hilbert) spaces; this claim is therefore not accurate. The proposed method is not essentially different from the kernel trick in the kernel SVM.

2. The numerical studies are not appropriately conducted. After reading the code in the supplemental file, there are several findings. (i) The standardization process includes both training and test data, which potentially leads to data leakage; (ii) There is no tuning of key hyperparameters and default values are used. SVMs are known to perform poorly without careful tuning. (iii) The presented results are based on only a single 80/20 split, which is not stable. (iv) AUC computed from hard class labels rather than probabilities/scores, which is essentially incorrect.

3. Unfair comparison: the method uses information from the test data (semi‑supervised/transductive), while the baselines are purely supervised.

4. The lemmas lack rigorous statements and proofs.

**Questions:**

1. It is commonly observed that kernel SVMs outperform linear SVMs; this is not the case in the reported results. I suspect this is due to the lack of tuning. Can the authors clarify?
2. The paper repeatedly refers to “N‑class SVM.” What is the performance on multi‑class data, and how is the method extended and evaluated in that setting?

---

### Meta-Review · Area_Chair_rQcD · 2026-01-06

**Summary:**

This paper proposes a covariance-adjusted SVM for non-Euclidean spaces by applying class-wise Cholesky whitening, reformulating the SVM optimization, and introducing an iterative algorithm to estimate population covariances, with experiments on five binary datasets.

Reviewers agree the topic is interesting and the empirical results are promising, but raise substantial concerns. The central claim that SVMs are valid only in Euclidean space is incorrect given kernel SVM theory. The theoretical development relies on informal lemmas without rigorous proofs. The proposed algorithm is heuristic, with unclear convergence and scalability properties.

Experimentally, evaluations are limited, comparisons are unfair due to transductive use of test data, and protocols are flawed (e.g., data leakage, single split, incorrect AUC computation). The author response did not sufficiently address these issues.

**Reviewer Concerns:**

Please see my summary.

**Reviewer Scores:**

It is difficult to say.  Overall, the authors provided some solid rebuttal, but it's a subjective judgement for the reviewer whether they would like to raise their score.

---

### Decision · Program_Chairs · 2026-01-26

Reject